# Attosecond coherent control of free-electron wave functions using semi-infinite light fields

G.M. Vanacore[1], I. Madan[1], G. Berruto[1], K. Wang[1,2], E. Pomarico[1], R.J. Lamb[3], D. McGrouther[3], I. Kaminer[1,2], B. Barwick[4], F. Javier García de Abajo [5,6] & F. Carbone[1]

Light–electron interaction is the seminal ingredient in free-electron lasers and dynamical investigation of matter. Pushing the coherent control of electrons by light to the attosecond timescale and below would enable unprecedented applications in quantum circuits and exploration of electronic motions and nuclear phenomena. Here we demonstrate attosecond coherent manipulation of a free-electron wave function, and show that it can be pushed down to the zeptosecond regime. We make a relativistic single-electron wavepacket interact in free-space with a semi-infinite light field generated by two light pulses reflected from a mirror and delayed by fractions of the optical cycle. The amplitude and phase of the resulting electron–state coherent oscillations are mapped in energy-momentum space via momentum-resolved ultrafast electron spectroscopy. The experimental results are in full agreement with our analytical theory, which predicts access to the zeptosecond timescale by adopting semi-infinite X-ray pulses.

[1] Institute of Physics, Laboratory for Ultrafast Microscopy and Electron Scattering (LUMES), École Polytechnique Fédérale de Lausanne, Station 6, 1015 Lausanne, Switzerland. [2] Department of Electrical Engineering, Technion—Israel Institute of Technology, Haifa 3200003, Israel. [3] SUPA, School of Physics and Astronomy, University of Glasgow, Glasgow G12 8QQ, UK. [4] Ripon College, 300 W. Seward St., Ripon, WI 54971, USA. [5] ICFO-Institut de Ciencies Fotoniques, The Barcelona Institute of Science and Technology, 08860 Castelldefels, Barcelona, Spain. [6] ICREA-Institució Catalana de Recerca i Estudis Avançats, Passeig Lluís Companys 23, 08010 Barcelona, Spain. These authors contributed equally: G.M. Vanacore, I. Madan. Correspondence and requests for materials should be addressed to F.J.G.d. A. (email: javier.garciadeabajo@nanophotonics.es) or to F.C. (email: fabrizio.carbone@epfl.ch)

The scattering of single photons by free-electrons is extremely weak, as quantified by the Thomson scattering cross-section, which for visible frequencies is of the order of $10^{-29}$ m². Additionally, direct photon absorption or emission by a free-space electron is forbidden due to energy-momentum mismatch. To circumvent these limitations and increase the probability of electron–photon interaction, a variety of methods have been devised[1]. For example, the Kapitza–Dirac effect involves a conceptually simple configuration in which an electron intersects a light grating produced by two counter-propagating light beams of the same frequency[2,3]. The interaction is then elastic and requires the electron to undergo an equal number of virtual photon absorption/stimulated-emission processes. When the absorbed and emitted photons differ in energy, the interaction results in frequency up- or down-conversion[4,5], which is the basis of undulator radiation and free-electron lasers[6–8].

A direct single-photon emission/absorption process can also bridge the energy-momentum mismatch if either the electrons are not free (for example, in photoemission from atoms/molecules[9] and solid surfaces[10]) or when a scattering structure generates evanescent light fields[11] in the vicinity of the interaction volume. Such an electron–photon–matter interaction creates optical field components with a frequency–momentum decomposition that lies outside the light cone, allowing emission/absorption to take place. This type of interaction, which is forbidden in free-space[12,13], is regularly exploited for generating radiation and for accelerating charged particles. Recently, it has also prompted the development of photon-induced near-field electron microscopy (PINEM)[11,14–16]. In PINEM, an energetic electron beam interacts with the evanescent near-fields surrounding an illuminated material structure. The interaction is particularly strong when the structure supports surface-plasmon polaritons (SPP) that are excited by short light pulses[17,18]. Optical near-fields then produce coherent splitting of the electron wave function in energy space, giving rise to Rabi oscillations among electron quantum states

separated by multiples of the photon energy[19]. The microscopic details of the process are encoded in the electron wave function, which can be revealed via ultrafast electron energy-loss spectroscopy (EELS) and controlled using suitable illumination schemes[20,21].

In this work, we adopt a more general method for controlling and manipulating the strength of electron–photon interaction. Instead of relying on localized near-fields (for example, plasmons), which inevitably depend on the intrinsic cross-section associated with the optical excitation of confined optical modes, we make use of a spatially abrupt interruption of the light field in free-space, also referred to as semi-infinite field[22–28] (see Fig. 1a). Such a boundary condition can be attained by sending the electrons through a light beam that intersects a refractor, an absorber, or more efficiently, a reflecting mirror along the optical path. When the light wave extends only over half-space, the energy-momentum conservation constraint is relaxed and electron–photon interaction can take place (see Fig. 1b and Supplementary Fig. 1) with an efficiency exceeding that produced by a resonant plasmonic nanostructure. Using this configuration, we have been able to simultaneously observe the quantized exchange of both energy and transverse momentum between a free-electron and a light wave, revealing the primary role of the quantum nature of the electron–light coupling at optical frequencies and above, and we provide a full description of the strength of interaction within parameter space.

Within this scenario, we demonstrate attosecond coherent control of the electron wave function by appropriately synthesizing a semi-infinite optical field using a sequence of two mutually phase-locked light pulses impinging on a mirror and delayed in time by fractions of the optical cycle (see schematics in Fig. 1c). The profile of the field resulting from such a temporal combination of pulses changes the energy and momentum of an electron as it traverses the interaction volume. The energy-momentum distribution of electron states is recorded as a function of the delay

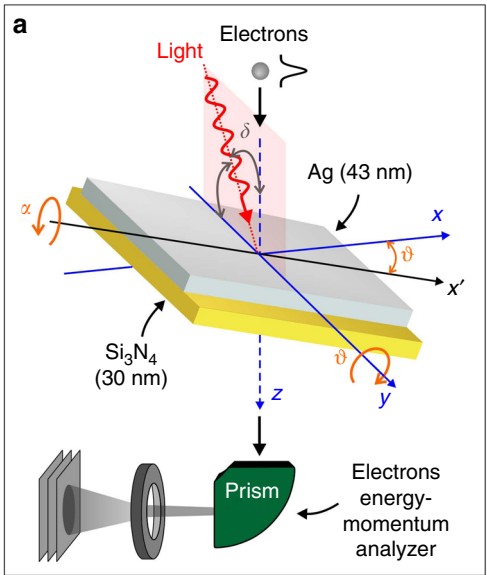
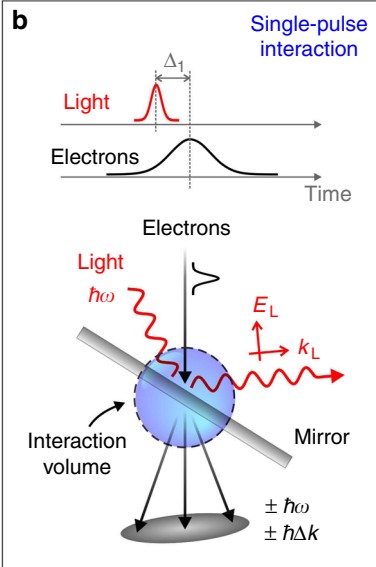
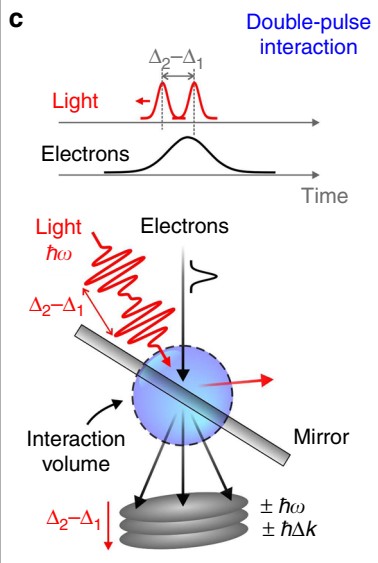

**Fig. 1** Experiment probing free-electron interaction with semi-infinite light fields. **a** Ultrashort 200 keV electron pulses travel along the $z$ axis and impinge on the surface of a Ag/Si₃N₄ thin bilayer, which is mounted on a double-tilt holder able to rotate around the $x$ (angle $\alpha$) and $y$ (tilt angle $\vartheta$) axes. Light propagates within the $y$–$z$ plane, incident with an angle $\delta \approx 4$–5° relative to the $z$ axis and then reflected from the Ag surface. The resulting electron–photon interaction is probed by monitoring electron energy-loss spectra as a function of geometrical parameters and light properties. **b** Description of the electron–light interaction here explored. The breaking of translational invariance produced by light reflection enables photon absorption or emission by the electron corresponding to a quantized energy and momentum exchange. **c** Description of the three-pulse experiment used for coherent modulation of the electron wave function. Electrons interact with an appropriately synthesized optical field distribution produced by two mutually phase-locked photon pulses whose relative phase is changed by varying their relative delay $\Delta_2 - \Delta_1$

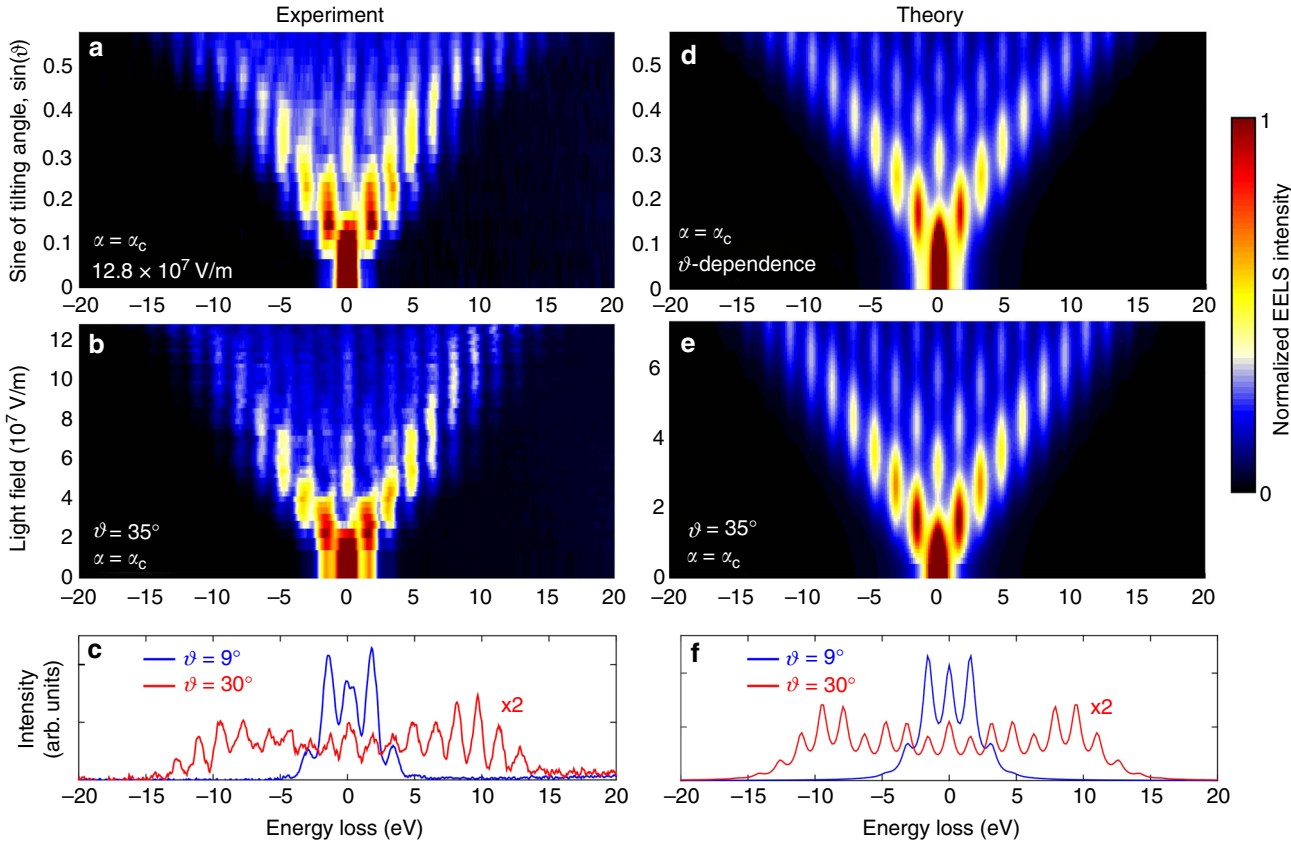

**Fig. 2** Energy exchange during electron–light interaction. **a** Sequence of measured EELS spectra (color map) plotted as a function of increasing angle $\vartheta$. We use p-polarized light (incident field along x axis), $\alpha = \alpha_C$, a peak field amplitude of $12.8 \times 10^7$ V/m, and light and electron pulse durations $\tau_L = 430$ fs and $\tau_e = 410$ fs. Sidebands at energies $\pm \ell \hbar \omega$ relative to the zero-loss peak (ZLP) are visible, where $\ell$ is the net number of exchanged photons. **b** Sequence of EELS spectra measured for increasing light field amplitude with fixed tilt angle $\vartheta = 35°$. **c** Spectra selected from **a**, measured at $\vartheta = 9°$ (blue curve) and $\vartheta = 30°$ (red curve), showing a strong redistribution of the electron density toward the high-energy sidebands for large tilt angle. **d–f** Simulated EELS spectra corresponding to the experimental conditions of **a–c** (see Methods for details of calculations)

between the two photon pulses via momentum-resolved fs-EELS performed in an ultrafast transmission electron microscope[29–31], revealing the light-induced modulation of both amplitude and phase of the electron wave function. In our scheme, the coherent control of the electron wave function is mediated by two temporally delayed pulses at the same position, in contrast with recently reported configurations[20,21,32–34] where the electron modulation was determined by the light interaction at different spatial positions along the electron pathway. This allows us to shift the interaction to the temporal domain instead of the spatial domain and thus taking full advantage of the intrinsic longitudinal coherence of the single-electron wave function while allowing us to explore other interesting scenarios such as the attosecond-nanometer modulation of plasmonic near-fields. Our experimental results are successfully described within a general theoretical framework for electron–light interaction, which is able to further predict the ability of this method to achieve coherent control over the electron wave function down to the zeptosecond regime using semi-infinite X-ray fields.

## Results

**Free-electron interaction with a semi-infinite light field.** The translational symmetry of a propagating electromagnetic wave is broken by refraction, absorption, or reflection at a material interface. In our study, we use a Ag thin film (43 nm) deposited on a $Si_3N_4$ membrane (30 nm) acting as a mirror. As

schematically depicted in Fig. 1a, the mirror is mounted on a double-tilt holder able to rotate around the x (angle $\alpha$) and y (tilting angle, $\vartheta$) axes. To demonstrate that electron–photon interaction can be strongly enhanced by the semi-infinite field effect, we display EELS spectra recorded as a function of laser field amplitude for a fixed orientation of the mirror (Fig. 2b), and as a function of mirror tilting angle $\vartheta$ for fixed field amplitude (Fig. 2a), using p-polarized light in all cases (incident field parallel to x axis). Following the interaction, the zero-loss peak (ZLP) at an energy $E_0 = 200$ keV is redistributed among sidebands at multiples of the incident photon energy $\pm \ell \hbar \omega$, corresponding to energy losses and gains by the electrons. At large values of both $\vartheta$ and the light field amplitude, the electron distribution is almost completely transferred toward high-energy spectral sidebands ($|\ell| \gg 1$), leaving a nearly depleted ZLP and revealing a high probability for multiphoton creation and annihilation.

The modulation of the EELS spectra is determined by the integral of the optical electric field amplitude $\mathcal{E}_z(z)$ along the electron-beam direction z. Following previous works[15,16,18], the strength of the electron–photon interaction can be quantified in terms of the parameter (see Methods)

$$\beta = (e/\hbar\omega) \int dz \, \mathcal{E}_z(z) e^{-i\omega z/v}. \quad (1)$$

In particular, the fraction of electrons transmitted in the $\ell$th

sideband is approximately given by the squared Bessel function

$$P_\ell = J_\ell^2(2|\beta|). \qquad (2)$$

The spectral distribution of the electron density can be thus changed either by tilting the mirror (Fig. 2b) or by increasing the laser power (Fig. 2c), producing quantitatively similar effects. Considering the large permittivity ($\approx -30 + 0.4i$) of silver at the employed photon energy ($\hbar\omega \approx 1.57$ eV) and the small optical skin depth ($\approx 11$ nm for $1/e$ decay in intensity) compared with the silver layer thickness, the mirror reflects >98% of the incident light. Thus, neglecting light penetration inside the material, the electric field along the electron path can be considered to be made of incident (i) and reflected (r) components as $\mathcal{E}_z(z) = \left( \mathcal{E}_z^i e^{ik_z^i z} + \mathcal{E}_z^r e^{-ik_z^r z} \right) \theta(-z)$, where the step function $\theta(-z)$ limits light propagation to the upper part of the mirror and $k_z^{i/r}$ is the projection of the incident/reflected light wave vector along $z$. Inserting this field into Eq. (1), we find

$$\beta \approx (ie/\hbar\omega)\left[ \frac{\mathcal{E}_z^i}{\omega/v - k_z^i} + \frac{\mathcal{E}_z^r}{\omega/v + k_z^r} \right], \qquad (3)$$

which makes the interaction strength finite and explicitly dependent on the field amplitude and tilting geometry. We further present in the Methods section a detailed analytical theory extended to deal with arbitrary pulse durations, two light pulses, and real material mirrors, used for comparison with the experimental results in the figures that follow. Nonetheless, Eq. (3) provides a satisfactory level of description that allows us to understand the data in simple terms, especially when the mirror is considered to be perfect (see Supplementary Figs. 2 and 3).

Because light and electron beams in our apparatus are not collinear, the interaction strength described by $\beta$ for p-polarized light vanishes only when the tilt angles are set to $\vartheta = 0°$ and $\alpha = \alpha_C = 12.9°$, in agreement with calculations based on the theory reported on Methods. This corresponds to the condition that the incident and reflected amplitudes almost completely cancel each other in Eq. (3), hence producing a negligible net effect (minimum $|\beta|$, see red curve in Supplementary Fig. 4). This result is also in agreement with the relation $\alpha_C = $

$\tan^{-1}[\sin\delta/(\cos\delta - v/c)]$ derived in the Supplementary Note 3 from Eq. (3) to yield $\beta = 0$ assuming a perfect mirror (blue curve in Supplementary Fig. 4). Likewise, $\beta$ cancels when the polarization is changed from p to s, a result that is clearly observed in polarization-dependent measurements (see Supplementary Fig. 5).

To extract quantitative information on the measurements presented in Fig. 2a–c, we perform the corresponding simulations shown in Fig. 2d–f for the energy distribution of a pulsed electron beam after impinging on an illuminated Ag/Si$_3$N$_4$ bilayer film, using the same layer thicknesses and geometrical arrangement as in the experiment. In particular, we consider p-polarized light incident with $\alpha$ fixed to the critical angle $\alpha_C$. Simulations are carried out incorporating realistic dielectric data for the involved materials (see Methods). The ratio of electron-to-light pulse durations $\tau_e/\tau_L \approx 410$ fs/430 fs $\approx 0.95$ is the same as estimated in experiment (see Methods), long enough to ensure large temporal overlap between the electron and light pulses, thus enhancing the probability of interaction. The agreement between experiment and theory is rather satisfactory. Similar conclusions are also obtained from measurements and simulations for small $\tau_L$ compared with $\tau_e$ (see Supplementary Figs. 3 and 6).

We remark that, in contrast to previous studies of electron–photon interactions[14], the effect here observed is primarily due to electrons coupling directly to the light waves rather than to the near-field created around a nanostructure. The kinematic mismatch in the electron–light coupling is remedied by the formation of semi-infinite light plane-waves (see Supplementary Fig. 1). As noted above, at a photon energy of $\approx 1.57$ eV the silver skin depth ($\approx 11$ nm) is much smaller than both the optical wavelength and the metal layer thickness, so the evanescent tail inside the Ag film gives a negligible contribution, as confirmed by direct comparison with perfect-mirror simulations based on Eq. (3) (see Supplementary Fig. 3).

**Quantized energy-momentum exchange in electron–photon coupling.** Energy exchanges between light and electrons should be also accompanied by momentum transfers along the direction parallel to the film, where translational invariance guarantees momentum conservation. Measuring such momentum exchanges

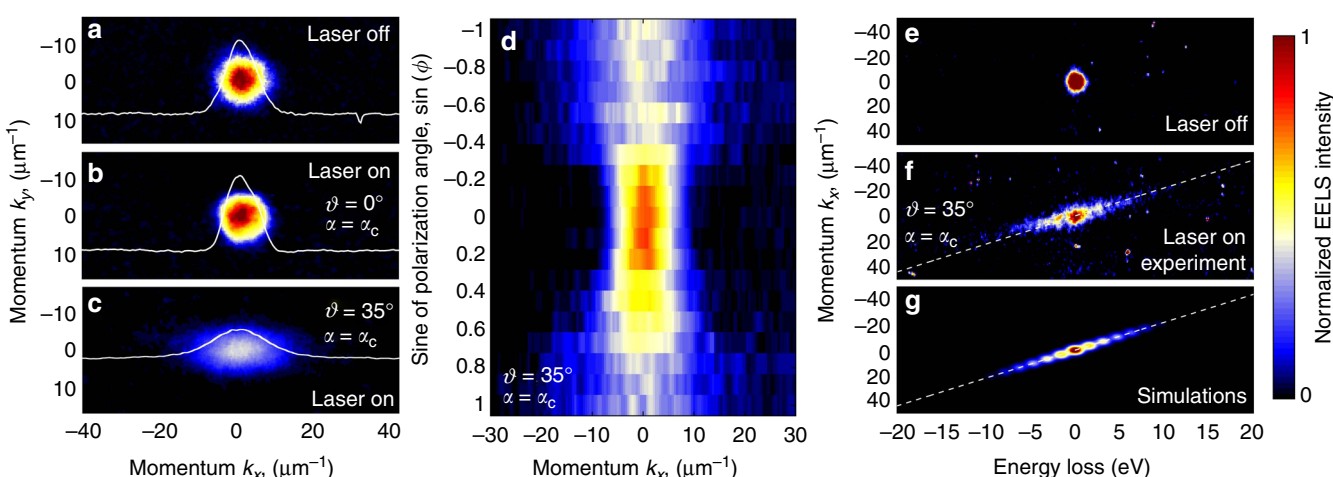

**Fig. 3** Momentum exchange during electron–light interaction. **a** Direct electron beam measured in the diffraction plane as a function of transversal momentum ($k_x, k_y$) when no light is applied. **b**, **c** Same as **a** under illumination with 560 fs laser pulses of $11.1 \times 10^7$ V/m peak field amplitude and $\alpha = \alpha_C$. The $\vartheta$ tilt angle is 0° in **b** and 35° in **c**. A clear streaking of the electron beam appears along the $k_x$ direction in **c** as a result of momentum exchanges between light and electrons. **d** Electron-beam profile along $k_x$ as a function of light polarization ($\sin\phi = 0$ for s-polarization and $\sin\phi = \pm 1$ for p-polarization). **e** Direct electron beam measured in the momentum-energy plane $k_x - E$ in the absence of optical illumination. **f**, **g** Measured (**f**) and simulated (**g**) momentum-energy maps for illumination under the conditions of **c**

is quite challenging because of the small induced electron deflection (only a few µrad), which demands high transverse coherence that we achieve by operating the microscope in high-dispersion diffraction mode. In Fig. 3a, we show the direct electron beam measured in the $k_x - k_y$ diffraction plane when no light is applied, whereas Fig. 3b and c shows the effect of light interaction for tilt angles $\vartheta = 0°$ and $\vartheta = 35°$, with fixed $\alpha = \alpha_C$. A clear streaking of the electron beam appears along the $k_x$ direction for $\vartheta = 35°$ as a result of the noted momentum exchange. As already observed in the electron energy spectra, the interaction vanishes at $\vartheta = 0$ and $\alpha = \alpha_C$ for p-polarization, resulting in zero transverse-momentum exchange. The physical origin of this behavior is well described by the analytical expressions in Eqs. (2) and (3), in which the electric field component along the $z$ axis modulates the interaction strength. This can be also experimentally probed by rotating the polarization of the light wave, which results in a corresponding modulation of the electron-beam streaking (see Fig. 3d). In our experiment, because the transverse coherence of the electrons is comparable with the light wavelength, a coherent quantized interaction between the electron wave function and the light wave is expected, rather than a purely classical deflection as mediated by the Lorentz force[32,33], which is generally negligible under excitation at optical photon frequencies and above.

This can be demonstrated by the simultaneous visualization of inelastic energy and transverse-momentum exchanges, which we directly map using the reciprocal-space imaging ability of the electron spectrometer in our microscope[17,29] (see Fig. 3e–g). The streaking of the electron beam occurs along a line in energy-momentum space with slope given by $q_{T,x}/\hbar\omega$, where $q_{T,x}$ is the transverse component of the transferred momentum along $x$, which in the limit of small angles $\delta$ and $\alpha$ admits the expression $q_{T,x} \approx (\omega/c)\cos\vartheta\sin\vartheta$ (see Supplementary Note 6 for the full derivation). For every photon absorption/emission event, the electron gains/loses a quantum of energy $\hbar\omega$ and momentum $\hbar q_{T,x}$ along $x$.

Such experiment yields a direct observation of the simultaneous quantized exchange of energy and transverse momentum between a propagating light wave and a free-electron, and shows the unique ability of our technique to map transient energy exchanges in momentum space. It could prompt the development of new microscopy methods in which the limitation imposed by EELS energy resolution is lifted for large momentum transfers, such as in the dynamic imaging of low-energy phonons. Furthermore, it demonstrates the ability of external electromagnetic fields to modulate the linear momentum, and potentially the angular momentum, of a free-electron in a dynamic way.

**Attosecond coherent control of an electron wave function.** These results provide a full characterization of electron–photon interaction at the mirror interface in energy-momentum space, which suggests using such interaction for the coherent manipulation of the electron wave function. We implement this idea by engineering the parameter $|\beta|$ (which can be thought of as a light-driven Rabi phase for transitions in the electron multilevel quantum ladder with $\hbar\omega$ energy spacings[19]) through a three-pulse experiment in which the electron interacts with a properly shaped field distribution consisting of a sequence of two mutually phase-locked photon pulses, delayed by time intervals $\Delta_1$ and $\Delta_2$ with respect to the electron pulse (see schematics in Fig. 1c and additional details in Methods). We change the relative phase between the two light pulses by varying $\Delta_2 - \Delta_1$ in steps of 500 attoseconds. The field distribution resulting from such a temporal combination of pulses is then used to coherently manipulate the energy-momentum distributions of the electrons.

A sequence of EELS spectra measured as a function of $\Delta_2 - \Delta_1$ is shown in Fig. 4a for $\vartheta = 35°$, $\alpha = \alpha_C$, $\tau_e \approx 350$ fs electron pulses, $\tau_L \approx 60$ fs optical pulses, a light field amplitude of $21.4 \times 10^7$ V/m per pulse, and delays $\Delta_1 = 0$ fs and $\Delta_2 \approx 100$–115 fs. The large values of $\Delta_2$ enable fine modulation of the optical phase while considerably reducing the intensity changes associated with light-pulse overlap. We observe periodic oscillations of the spectral sidebands with a period $\approx 2.6$ fs equal to the optical cycle $2\pi/\omega$. Detailed inspection of the EELS spectra for two different delays ($\Delta_2 = 109$ fs and 110.5 fs in Fig. 4b, corresponding to the horizontal dashed lines in Fig. 4a) reveals radically different distributions of the sidebands relative to the ZLP, which are further quantified in Fig. 4c by plotting the $\ell = 9$ and $\ell = 14$ features as a function of $\Delta_2 - \Delta_1$. We observe significant intensity oscillations with a period of $\approx 2.6$ fs and a well-defined $\sim \pi$ relative phase shift. We remark once more that measurements shown in Fig. 4a–c are well reproduced by our analytical simulations for two light pulses (see Methods) plotted in Fig. 4d–f, respectively. As described in detail in Supplementary Note 7, where we have included several control experiments, additional calculations, and further considerations on the intrinsic temporal coherence of the single-electron wave function, we demonstrate that this effect cannot be assimilated to a simple intensity variation of the impinging light, which stays at the $\sim\pm 5 \times 10^{-2}$ level, and neither to an incoherent interaction between the electrons and the two temporally delayed pulses. In fact, in the latter case the modulation of the energy spectrum, and especially of the high-energy sidebands, would be only determined by the 5%-optical interference and would be quantitatively in a similar range, in contrast with the experimental observations.

The measured oscillatory behavior is indicative of a continuous redistribution within the quantum electron-population ladder, periodically transferred back and forth between high- and low-energy levels. Such an effect is the result of coherent modulation of the electron wave function via the coherent constructive and destructive modulation of $|\beta|$ when changing the relative phase between the two driving optical pulses. The time-Fourier transform of the maps in Fig. 4a and d, presented in Fig. 5a and c, gives access to the spectral distribution within the quantum ladder at the modulation frequency $2\pi/(2.6$ fs$) \approx 385$ THz. The amplitude and phase of such a modulation, shown in Fig. 5b and d, provide a complete picture of the optically manipulated electron wave function resolved for each electron energy level.

The coherent control of ultrafast electron beams has recently attracted much attention for its potential application in ultrashort (attosecond) electron sources, as well as electron imaging and spectroscopy. While semi-infinite light beams have been used for the temporal streaking and compression of electron pulses[21,32–34], here we demonstrate the simultaneous quantized exchange of energy and transverse momentum between electrons and light, which is the dominant mechanism at optical frequencies and above, and we provide a direct measurement of the strength of this quantum coherent interaction for controlling the electron energy-momentum distribution. In our experiments, we synthesize a semi-infinite temporally modulated field distribution (obtained by a sequence of two mutually phase-locked light pulses impinging on a mirror) to demonstrate coherent modulation of the electron wave function. A schematic representation of such modulation is shown in Fig. 5e, where snapshots of the strong electron density redistribution in both energy and momentum, as observed experimentally and calculated theoretically, are presented for different values of the optical phase shift of the synthesized optical field distribution. This approach allows us to develop additional capabilities of coherent control of free-electrons beyond similar configurations adopted so far, where the electron wave function modulation is determined by the light

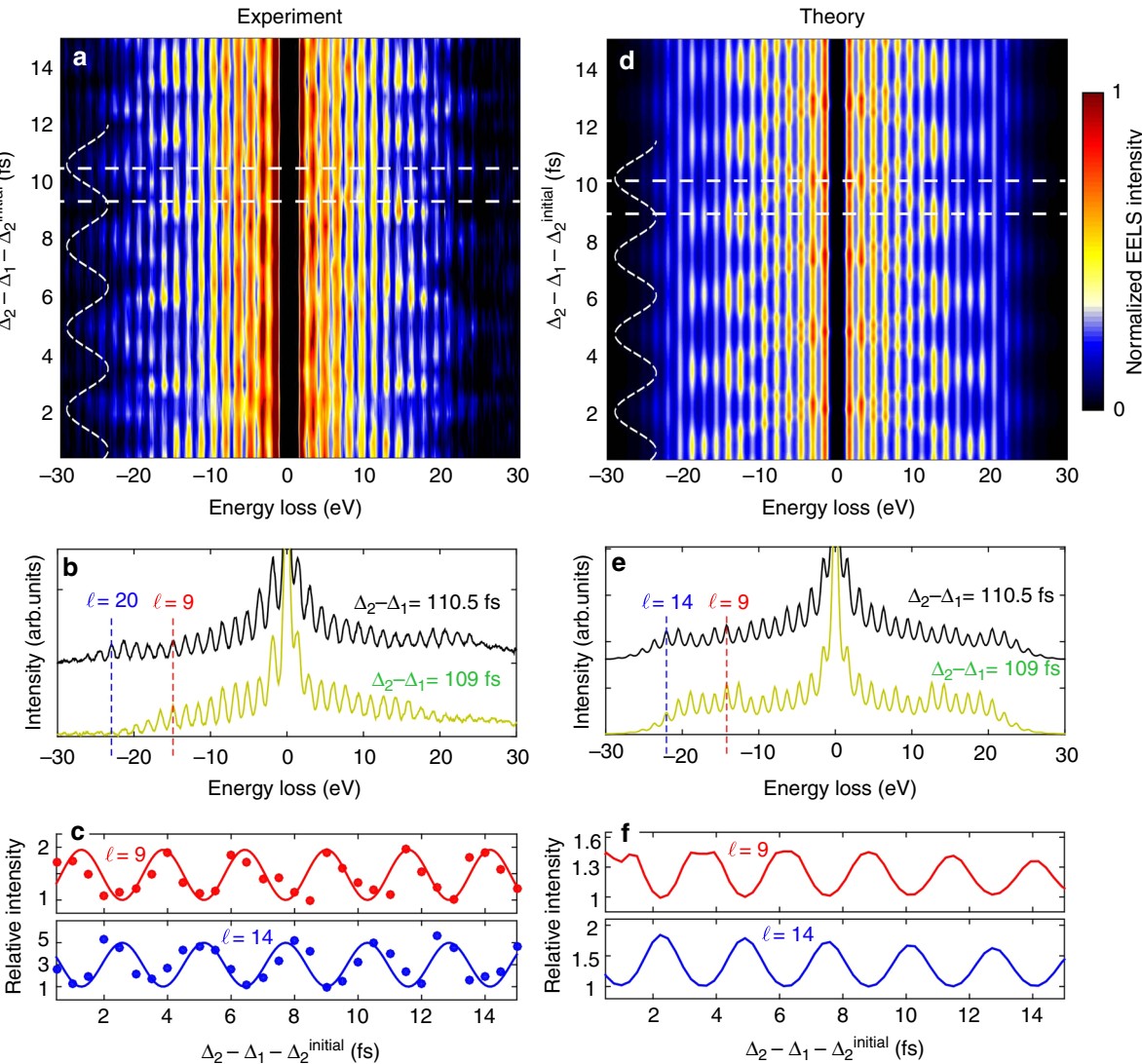

**Fig. 4** Attosecond coherent control of free-electrons. The electron beam interacts with a semi-infinite temporally modulated optical field distribution produced by a sequence of two mutually phase-locked light pulses impinging on the mirror. **a** Measured EELS spectra as a function of relative delay $\Delta_2 - \Delta_1$ between the two optical pulses. The tilt angles are $\vartheta = 35°$ and $\alpha = \alpha_C$, the optical pulses are 60 fs long with a peak field amplitude of $21.4 \times 10^7$ V/m each, and the delays are $\Delta_1 = 0$ and $\Delta_2 \approx 100$–115 fs, with $\Delta_2^{initial} = 100$ fs. **b** EELS spectra taken at two different time delays (marked by horizontal dashed lines in **a**). **c** Relative intensity (full circles) of the $\ell = 9$ and $\ell = 14$ sidebands plotted as a function of time delay between the two optical pulses, exhibiting a periodic modulation of period $\approx 2.6$ fs (equal to the optical cycle $2\pi/\omega$) and a relative $\pi$ phase shift. Solid curves are least-square fits to the data. **d**–**f** Simulated EELS spectra and resulting intensity change corresponding to the experimental conditions of **a**–**c** (see Methods for details of calculations)

interaction at different spatial positions along the electron pathway[20,21,32–34]. In our scheme, the adoption of two temporally separated semi-infinite light fields on one flat and homogeneous thin layer allows us to employ a simpler experimental geometry and shift the two interactions temporally instead of spatially, thus taking full advantage of the intrinsic longitudinal coherence of the single-electron wave function.

**Attosecond-nanometer control of plasmonic near-fields.** Overall, this experiment–theory framework is general and allows describing other interesting scenarios, such as the phase-controlled combination of the interaction arising from both semi-infinite light fields and plasmon polaritons propagating on a metal film. This is illustrated by measurements presented in Supplementary Fig. 11, with SPPs optically generated at the edge of a linear nanocavity carved in the Ag layer. The interference between the traveling plasmon wave and the semi-infinite light field creates a standing wave distribution sampled by the

electrons, which allows us to produce a snapshot of the propagating SPP in real-space. By using the two-pulse scheme described above, coherent control of these plasmonic near-fields can be achieved at attosecond-nanometer scale. This is demonstrated by using a nano-fabricated plasmonic Fabry–Perot (FP) resonator (see Supplementary Fig. 12a), while simultaneously adopting an experimental geometry that cancels the interaction with the semi-infinite field ($\alpha = \alpha_C$ and $\vartheta = 0°$), allowing the resonant plasmon modes of the FP to be solely imaged (see Fig. 6a). Varying the delay between the two optical pulses in steps of 334 as (see scheme in Supplementary Fig. 12b) allows us to control the relative phase between the optically excited plasmons, resulting in a time-dependent sequence of constructive and destructive interference between them (see Fig. 6b, c). The plasmonic coherent control experiments presented here, which would have been unfeasible in other schemes involving spatially separated interactions, offer the unique opportunity to perform time-domain spectro-microscopy of plasmon resonances, where the

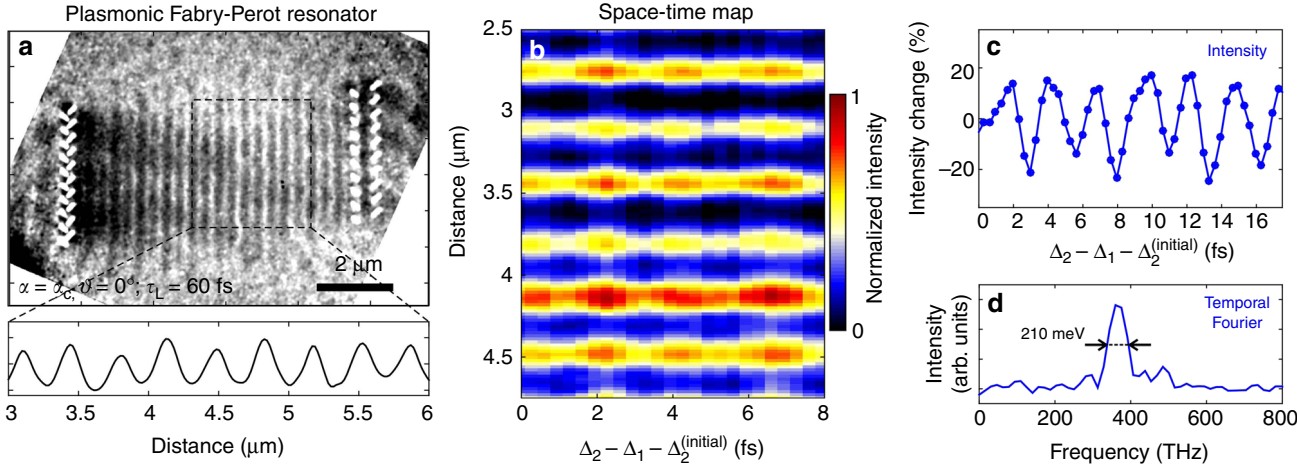

**Fig. 5** Amplitude and phase modulation of the electron wave function. **a** Two-dimensional Fourier transform of the energy-time map plotted in Fig. 4a. **b** Complex spectral distribution of the electron wave function amplitude (bottom) and phase (top), extracted at the modulation frequency of $2\pi/(2.6\,\text{fs}) \approx$ 385 THz. **c**, **d** Two-dimensional Fourier transform extracted from the calculated energy-time map plotted in Fig. 4d. **e** Schematic representation of electron wave function modulation, showing snapshots of the strong energy-momentum electron density redistribution for different values of the phase shift between the two optical pulses

**Fig. 6** Attosecond-nanometer modulation of plasmonic near-fields. **a** Energy-filtered image of the plasmonic interference pattern created in the designed plasmonic Fabry–Perot resonator. The tilt angles are set to $\vartheta = 0°$ and $\alpha = 12.9°$, while the optical pulses are 60 fs long. **b** Spatiotemporal mapping of the plasmonic coherent control at attosecond-nanometer scale. **c** Measured temporal modulation of the plasmon-mode amplitude. **d** Fourier transform of the measured temporal trace

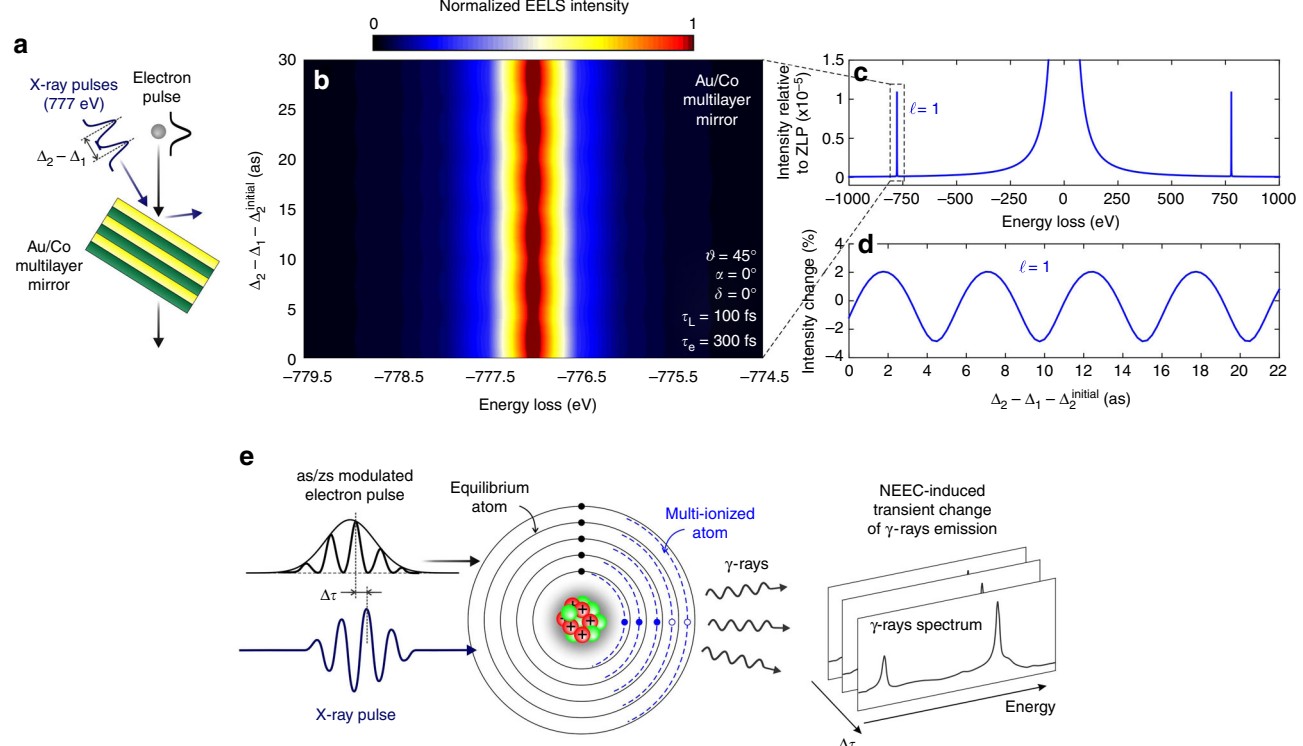

**Fig. 7** Zeptosecond coherent control of free-electrons. **a** An electron beam interacts with a semi-infinite temporally modulated X-ray field (777 eV photon energy) produced by a sequence of two mutually phase-locked pulses partially reflected by a Au/Co multilayer. **b** Calculated EELS spectra as a function of relative delay $\Delta_2 - \Delta_1$ between the two X-ray pulses. The tilt angles are set to $\vartheta = 45°$ and $\alpha = 0°$, while the X-ray pulses are 100 fs long with a peak field amplitude of $9.4 \times 10^9$ V/m per pulse. Simulations are performed within a 30-attosecond window starting from $\Delta_2^{initial} = 150$ fs ($\Delta_1 = 0$). **c** Calculated EELS spectrum showing the electron sidebands at energies of ±777 eV with respect to the ZLP. **d** Relative intensity change of the $\ell = 1$ sideband plotted as a function of time delay between the two X-ray pulses, exhibiting a periodic modulation of period ≈5.3 as (equal to the X-ray cycle) and an intensity change rate of ≈1% per 511 zs. **e** Schematic description of a gedanken-experiment for external control of nuclear excitations. A train of zs electron pulses is overlapped with the optical cycle of an X-ray pulse. Their relative delay $\Delta\tau$ is externally varied while monitoring the change in the $\gamma$-ray emission as a result of multiple ionization/electron capture events. The atomic levels for the equilibrium and ionized atom are pictorially represented

energy resolution is obtained via Fourier transform of the temporal traces (see Fig. 6d) and is not limited by the overall electron energy-loss resolution (sub-eV at best, see Supplementary Fig. 10). In a complementary frequency-domain approach[35], the spectral response of the resonance was obtained by using a single optical pulse with a tunable wavelength and a resolution determined by the 20 meV laser linewidth.

**Zeptosecond coherent control of an electron wave function**. As described above, when the electron scattering cannot be assisted by the plasmonic near-fields, a refracting, absorbing, or reflecting interface can be used for mediating the electron–light interaction. A particularly appealing consequence of such condition consists in the possibility of controlling the electron wave function using photons of different energies, not restricted by the ability of materials to support localized plasmon resonances, but solely determined by the quality of the mirror surface at a specific frequency. Using high-energy photons all the way to the X-ray regime, our methodology would then allow us to control the electron wave function down to the zeptosecond timescale[36]. To verify the feasibility of this idea, we have designed a multilayer mirror composed of 30 layers of 1.6-nm-thick cobalt spaced by 1-nm-thick gold (total thickness is 78 nm), still transparent for 200 KeV electrons and capable of reflecting around 35% of 777 eV light at an angle of incidence of 45° (see Supplementary Fig. 13). This type of mirror is routinely used in X-ray facilities[37], and combined with commonly employed TEM sample preparation techniques, such as ion-milling and FIB machining, it can be

fabricated in form of electron transparent lamellas. We then simulated a three-pulse experiment with two 100 fs, 777 eV, 50 TW/cm² X-ray pulses, similar to what is currently available from free-electron lasers[38], impinging on the multilayer along the same direction as a 300 fs electron pulse (see schematics in Fig. 7a). We carry out simulations within a 30-attosecond window starting from an initial delay $\Delta_2 - \Delta_1 = 150$ fs. Electron sidebands are clearly discernable at energies of ±777 eV relative to the ZLP (see Fig. 7c), originating in the same electron–ladder interaction as observed for near-infrared light. From our calculation, we infer that the sideband $\ell = 1$ has a relative intensity with respect to the ZLP of about $10^{-5}$. For a repetition rate of 300 kHz, such as used in the LCLS-II FEL at SLAC, this translates to about 3 electrons/s in a single channel of the detector, whose measurement can be done using commercially available highly sensitive direct detector cameras. The resulting EELS spectrum as a function of the delay between the two X-ray pulses is displayed in Fig. 7b, and the relative intensity change for the first sideband is shown in Fig. 7d, revealing a clear modulation by the optical cycle of the X-ray pulse (≈5.3 as) and an intensity change rate of ≈1% per 511 zs. It is worth noting that in our scheme phase fluctuations of the X-ray beam, whose main effect is the generation of a temporal jitter of few femtoseconds[39] between consecutive pulses, do not represent an issue. This is because our experiment uses two photon pulses originating from the same photon pulse by means of an interferometer, and thus, the two X-ray pulses will be intrinsically phase-locked with an inherently zero jitter between them. Coherent manipulation of the electron wave function can be thus

pushed to the zeptosecond regime using currently existing technology within our electron–light interaction scheme. Access to such timescales may open interesting perspectives for the observation of intramolecular electronic motions[40] and nuclear processes such as fission, quasifission, and fusion[41].

**External control of nuclear excitations**. Very recently, nuclear excitation by electron capture (NEEC) has been experimentally demonstrated[42]. In such a process, an electron is captured by an ionized atom while simultaneously inducing the excitation of the nucleus. In this experimental design, ions were produced by stripping electrons away from an atomic beam going through a thin foil. The electronic levels of the ionized atoms were redistributed with respect to the equilibrium atoms, so that electrons interacting with them randomly sampled one of these configurations.

On a different approach, multiple atomic ionization can also be produced by interaction with ultrashort intense laser pulses. In ref. [43], a surprising resonant effect is reported when tuning the energy of the ionizing X-ray laser pulse. During this process, atomic levels transit several intermediate configurations whose lifetimes lie in the range between few zeptoseconds and few attoseconds.

In this scenario, we propose that by synchronizing the carrier of a properly tuned X-ray pulse with ultrashort electron pulses at the attosecond or zeptosecond level, the nuclear excitation can be controlled coherently with an ad hoc removal/insertion of electrons from and into the atom. A scheme of this concept is displayed in Fig. 7e. A train of zs electron pulses is synchronized to the optical cycle of an X-ray pulse. By varying their relative delay time $\Delta\tau$, different out-of-equilibrium configurations of the ionized atoms may be sampled in a push/pull-like approach. Given the degrees of freedom that such an experiment can provide in choosing both the electron and X-ray energies, their relative timing, light polarization and intensity, we expect that interesting resonance effects can be discovered in the excitation of the nuclei.

Controlling nuclear phenomena via external parameters is an extremely interesting perspective. Ideally, one would like to induce instabilities in an otherwise stable or metastable nucleus to prompt energy-producing decays, or to generate radiation. However, accessing nuclei is difficult and energetically costly because of the protective shell of electrons surrounding it. Thus, external parameters such as pressure, magnetic field or chemical environment have little or no effect on decay rates and nuclear properties in general. Our scheme would offer a further perspective for the control of nuclear reactions with potential implications in various fields, from fundamental physics to energy-related applications.

## Methods

**Materials and experiment**. A sketch of our experiment is depicted in Fig. 1a. We used an ultrafast transmission electron microscope (a detailed description can be found in ref. [29]) to focus femtosecond electrons and light pulsed beams on an optically thick mirror. The mirror was thin enough to transmit the electrons while producing large light reflection. Specifically, it was made of a 43 nm-thick (±5 nm) silver thin film sputtered on a 30 nm $Si_3N_4$ membrane placed on a Si support with a $80 \times 80\ \mu m^2$ window, which was in turn mounted on a double-tilt sample holder that ensured rotation around the $x$ (angle $\alpha$) and $y$ (angle $\vartheta$) axes over a ±35° range. Electron pulses were generated by photoemission from a UV-irradiated $LaB_6$ cathode, accelerated to an energy $E_0 = 200$ keV along the $z$ axis, and focused on the specimen surface. The mirror was simultaneously illuminated with femtosecond laser pulses of $\hbar\omega = 1.57$ eV central energy and variable duration, intensity, and polarization. The light pulses were focused on the sample surface (spot size of ~58 μm FWHM). The light propagation direction lied within the $y$–$z$ plane and formed an angle $\delta \sim 4$–5° with the $z$ axis, as shown in Fig. 1a. The delay between electrons and photons was varied via a computer-controlled delay line. For the three-pulse experiment, we implemented a Michelson interferometer along the

optical path of the infrared beam, incorporating a computer-controlled variable delay stage on one arm.

The transmission electron microscope was equipped with EELS capabilities, coupled to real-space and reciprocal-space imaging. Energy-resolved spectra were acquired using a Gatan imaging filter (GIF) camera operated with a 0.05 eV-per-channel dispersion setting and typical exposure times of the CCD sensor from 30 to 60 s. Multiple photon absorption and emission events experienced by the electrons were analyzed as a function of relative beam-mirror orientations by recording EELS spectra and diffraction patterns in high-dispersion-diffraction mode. During post-acquisition analysis, the EELS spectra were aligned based on their ZLP positions using a differential-based maximum intensity alignment algorithm.

Special care was taken in modulating and evaluating the temporal width of the light and electron pulses. We varied the duration of the optical pulses by modifying the temporal chirp of the laser amplifier output using a pair of tunable glass prisms. An infrared auto-correlator was used for measuring the duration of the infrared pulses. For electrons, the pulse duration was estimated by measuring the electron–photon cross-correlation as obtained by monitoring the EELS spectra as a function of the delay time between electrons and the infrared light. In the low-excitation regime, the measured temporal width of the $\ell$th sideband is roughly $\tau_\ell \approx \sqrt{(\tau_e^2 + (\tau_L^2)/\ell)}$ (that is, the convolution of electron and optical pulses[44] of durations $\tau_e$ and $\tau_L$, respectively). For infrared pulses with $\tau_L = 60$, 175, and 430 fs FWHM, we derived electron pulse durations $\tau_e = 350$, 395, and 410 fs FWHM, respectively (<5% estimated error).

**Theory of ultrafast electron–light interaction**. Following previous works[15,16,18], we describe an electron wavepacket exposed to an optical field through the Schrödinger equation $(H_0 + H_1)\psi = i\hbar\partial\psi/\partial t$, where $\psi(\mathbf{r}, t)$ is the electron wave function, $H_0$ is the free-space Hamiltonian, and $H_1 = (-ie\hbar/m_e c)\mathbf{A}(\mathbf{r}, t) \cdot \nabla$ represents the minimal-coupling interaction involving the optical vector potential $\mathbf{A}(\mathbf{r}, t)$ in a gauge in which the scalar potential and $\nabla \cdot \mathbf{A}$ are both zero. We consider an expansion of the components $e^{i(\mathbf{k}\cdot\mathbf{r} - E_\mathbf{k}t/\hbar)}$ of the electron wave function in terms of momentum $\hbar\mathbf{k}$ piled near a central value $\hbar\mathbf{k}_0$ with $k_0 = \hbar^{-1}\sqrt{(2m_e E_0)(1 + E_0/2m_e c^2)}$, corresponding to an electron kinetic energy $E_0$. Each of these components is an eigenstate of $H_0$ with energy $E_\mathbf{k} \approx E_0 + \hbar\mathbf{v} \cdot (\mathbf{k} - \mathbf{k}_0)$, where $\mathbf{v} = (\hbar\mathbf{k}_0/m_e)/(1 + E_0/m_e c^2)$ is the central electron velocity. This approximation is valid for small momentum spread (that is, $|\mathbf{k} - \mathbf{k}_0| \ll k_0$). Under these conditions, we can also approximate $H_0 \approx E_0 - \hbar\mathbf{v} \cdot (i\nabla + \mathbf{k}_0)$, as well as $\nabla \approx i\mathbf{k}_0$ in $H_1$. Now, it is convenient to separate the fast evolution of the wave function imposed by the central-momentum component as $\psi(\mathbf{r}, t) = e^{i(\mathbf{k}_0\cdot\mathbf{r} - E_0 t/\hbar)}\phi(\mathbf{r}, t)$, where $\phi(\mathbf{r}, t)$ then displays a slower dynamics. Putting these elements together, the Schrödinger equation reduces to

$$(\mathbf{v} \cdot \nabla + \partial/\partial t)\phi = \frac{-ie\mathbf{v}}{\hbar c} \cdot \mathbf{A}\phi,$$

which admits the rigorous solution

$$\phi(\mathbf{r}, t) = \phi_0(\mathbf{r} - \mathbf{v}t)\exp\left[\frac{-ie\mathbf{v}}{\hbar c} \cdot \int_{-\infty}^{t} dt'\, \mathbf{A}(\mathbf{r} + \mathbf{v}t' - \mathbf{v}t, t')\right]. \quad (4)$$

Here, $\phi_0(\mathbf{r} - \mathbf{v}t)$ is the electron wave function before interaction with the optical field. In practice, we consider illumination by an optical pulse with a narrow spectral distribution centered around a frequency $\omega$, so the vector potential can be approximated as $\mathbf{A}(\mathbf{r}, t) \approx (-ic/\omega)\vec{\mathcal{E}}_0(\mathbf{r}, t)e^{-i\omega t}$ + c.c., where the electric field amplitude $\vec{\mathcal{E}}_0(\mathbf{r}, t)$ describes a slowly varying pulse envelope that changes negligibly over an optical period. Inserting this expression into Eq. (4), we find the solution $\phi(\mathbf{r}, t) = \phi_0(\mathbf{r} - \mathbf{v}t)e^{-\mathcal{B}+\mathcal{B}^*}$, where $\mathcal{B}(\mathbf{r}, t) = \frac{e\mathbf{v}}{\hbar\omega} \cdot \int_{-\infty}^{t} dt'\, \vec{\mathcal{E}}_0(\mathbf{r} + \mathbf{v}t' - \mathbf{v}t, t')e^{-i\omega t'}$. Finally, using the Jacobi-Anger expansion $e^{iu\sin\varphi} = \sum_{\ell=-\infty}^{\infty} J_\ell(u)e^{i\ell\varphi}$ (see Eq. (9.1.41) of ref. [45]) with $|u| = 2|\mathcal{B}|$ and $\varphi = \arg\{-\mathcal{B}\}$, we obtain $\phi(\mathbf{r}, t) = \phi_0(\mathbf{r} - \mathbf{v}t)\sum_{\ell=-\infty}^{\infty} J_\ell(2|\mathcal{B}|)e^{i\ell\arg\{-\mathcal{B}\}}$. This expression has general applicability under the assumptions of small energy spread in both electron and optical pulses.

For monochromatic light (that is, when $\vec{\mathcal{E}}_0(\mathbf{r})$ depends only on position), considering without loss of generality $\mathbf{v}$ along $\hat{\mathbf{z}}$, we find $\mathcal{B} = \beta(\mathbf{r})e^{-i\omega(z/v - t)}$ with

$$\beta(\mathbf{r}) = \frac{e}{\hbar\omega}\int_{-\infty}^{z} dz'\, \mathcal{E}_{0z}(x, y, z')e^{-i\omega z'/v}, \quad (5)$$

and the electron wave function then becomes

$$\phi(\mathbf{r}, t) = \phi_0(\mathbf{r} - \mathbf{v}t)\sum_{\ell=-\infty}^{\infty} J_\ell(2|\beta|)e^{i\ell\arg\{-\beta\} + i\ell\omega(z/v - t)}, \quad (6)$$

where the last term in the exponential shows a change in the energy and momentum of the $\ell$ wave function component given by $\ell\hbar\omega$ and $\ell\hbar\omega/v$.

For a Gaussian light pulse $\vec{\mathcal{E}}_0(\mathbf{r}, t) = \vec{\mathcal{E}}_0(\mathbf{r})e^{-t^2/\sigma_L^2}$, corresponding to a FWHM-intensity duration $\tau_L = (\sqrt{2\log 2})\sigma_L \approx 1.18\sigma_L$, under the assumption that the time needed by the electron to cross the interaction region is small compared with $\sigma_L$, we recover the result of Eq. (6) with $\beta$ (Eq. (5)) replaced by $e^{-(z/v - t)^2/\sigma_L^2}\beta$.

We now calculate the electron probability at the detector as the integral $\int d^3\mathbf{r}|\phi(\mathbf{r}, t)|^2$ for a large time $t$. Assuming a Gaussian electron pulse $\phi_0(\mathbf{r} - \mathbf{v}t) \propto e^{-(t-z/v-\Delta_1)^2/\sigma_e^2}$ normalized to one electron $\left(\int d^3\mathbf{r}|\phi_0|^2 = 1\right)$, with FWHM-intensity duration $\tau_e = \left(\sqrt{2\log 2}\right)\sigma_e$, and a delay $\Delta_1$ relative to the light pulse, we find the probability that the electron has exchanged a net number of photons $\ell$ to be

$$P_\ell = \sqrt{\frac{2}{\pi}}\frac{1}{\sigma_e}\int dt\, e^{-2t^2/\sigma_e^2} J_\ell^2\left(2|\beta|e^{-(t+\Delta_1)^2/\sigma_L^2}\right), \quad (7)$$

with $\beta$ evaluated in the $z \to \infty$ limit of Eq. (5). From the identity[45] $\sum_\ell J_\ell^2(u) = 1$, we reassuringly obtain $\sum_\ell P_\ell = 1$. In the derivation of this expression, we have assumed that different $\ell$ electron channels have well separated energies, a condition that is guaranteed by the assumption of small energy spread in both pulses (that is, $E_0\sigma_e \gg \hbar$ and $\omega\sigma_L \gg 1$). Finally, using the Taylor expansion $J_\ell(u) = \sum_{j=0}^{\infty}(-1)^j(u/2)^{\ell+2j}/j!(\ell+j)!$ for the Bessel functions[45], the time integral in Eq. (7) can be readily performed term by term to yield

$$P_\ell = \sum_{j=0}^{\infty}\sum_{j'=0}^{\infty} C_{\ell j}C_{\ell j'}\frac{1}{\sqrt{\lambda}}e^{-2n\left(\Delta_1^2/\sigma_L^2\right)/\lambda}, \quad (8)$$

where $n = \ell + j + j'$, $\lambda = 1 + n(\sigma_e/\sigma_L)^2$, and $C_{\ell j} = (-1)^j|\beta|^{\ell+2j}/j!(\ell+j)!$. In the monochromatic limit $\sigma_L \gg \sigma_e$, we trivially obtain $P_\ell = J_\ell^2(2|\beta|)$ (that is, Eq. (2)).

Under illumination with two identical light pulses delayed by $\Delta_i$ ($i = 1, 2$) relative to the electron and with their amplitudes scaled by real factors $A_i$, a similar analysis can be carried out, starting by expressing $\beta$ as the sum of two contributions (one per light pulse). Under the assumptions stated above, we obtain for the probability an expression similar to Eq. (7),

$$P_\ell = \sqrt{\frac{2}{\pi}}\frac{1}{\sigma_e}\int dt\, e^{-2t^2/\sigma_e^2} J_\ell^2\left(2\left|\beta\sum_{i=1,2}A_i e^{-i\omega\Delta_i}e^{-(t+\Delta_i)^2/\sigma_L^2}\right|\right). \quad (9)$$

We then replace the Bessel function by its Taylor expansion and use the Newton binomial expansion to work out the powers of the $i$ sum. Each term in the resulting expression has a time dependence fully contained in a single exponential with an argument having terms in $t$ and $t^2$, which we integrate analytically. After some tedious but straightforward algebra, we find the result

$$P_\ell = \sum_{j=0}^{\infty}\sum_{j'=0}^{\infty}\sum_{s=0}^{n}\sum_{s'=0}^{n} C_{\ell j}C_{\ell j'}\binom{n}{s}\binom{n}{s'}A_1^{2n-s-s'}A_2^{s+s'}\cos[(s-s')\omega(\Delta_2-\Delta_1)]$$
$$\times \frac{1}{\sqrt{\lambda}}e^{-2n\left(\Delta_{12}^2/\sigma_L^2\right)/\lambda}\,e^{-[1-(s-s')/2n](s-s')(\Delta_2-\Delta_1)^2/\sigma_L^2}, \quad (10)$$

where $\Delta_{12} = [(2n-s-s')\Delta_1 + (s+s')\Delta_2]/2n$. While Eq. (10) is convenient for the calculation of probabilities, Eq. (9) delivers a clearer physical picture: the argument of the Bessel function incorporates the coherent superposition of the two pulses, in which the shared spatial dependence affects $\beta$ (Eq. (5)), while the temporal dependence stemming from $\mathcal{B}$ is captured by the $i$ sum. In particular, a trivial constructive (destructive) interference takes place in the limit of long pulses if $A_1 = A_2$ and $\Delta_1 - \Delta_2$ is a multiple (half-multiple) of the optical period. For finite electron pulses, this interference is more involved, as it is intermingled with different components of the electron wave function along the time integral (such as captured by the factor $e^{-2n\left(\Delta_{12}^2/\sigma_L^2\right)/\lambda}$ in Eq. (10)).

The above theory includes the temporal span of the electron function through the Gaussian FHWM parameter $\sigma_e$. Incidentally, we have also generalized this result to include an incoherent temporal broadening of the electrons through a Gaussian temporal distribution of the electron wavepacket center with a FWHM $\sigma_{e,inc}$; this analysis rigorously leads to the same expressions as above, but with $\sigma_e$ substituted by $\sqrt{\sigma_e^2 + \sigma_{e,inc}^2}$, thus indicating that within the assumptions of the present model the PINEM spectra depend on coherent (wave function temporal span) and incoherent (different times of arrival) electron broadening through a single parameter that coincides with the convolution of two Gaussians of durations $\sigma_e$ and $\sigma_{e,inc}$.

In our numerical simulations, we use Eqs. (8) and (10) with the electric field obtained by a standard transfer-matrix approach for a bilayer formed by Ag and $Si_3N_4$, with the permittivities of these materials taken as[46] $-30.3 + 0.39i$ and[47] $4.04$, respectively. Calculations for X-ray pulses at 777 eV photon energy are performed for multilayers of Au and Co, described by their permittivities $0.97 + 0.014i$ and $1.01 + 0.0014i$, respectively. Light amplitudes in the simulations are reduced by a factor of 1.7 with respect to the experimental estimates. This factor, which provides the best theory-experiment fit, is presumably originating in unaccounted losses along the optical path of the laser beam, especially when the light enters within the electron microscope before reaching the sample. This might possibly be due to contamination of the metallic mirror inside our TEM or to partial clipping of the light beam by the inner structure of the magnetic lens. Also, because the estimate of the beam diameter at the sample is carried out through indirect methods, it can easily be underestimated. If we assume a 30% underestimation of the diameter, we get a factor 1.5 in the electric field amplitude, not far from 1.7.

**Data availability**. The data that support the findings of this study are available from the corresponding authors upon reasonable request.

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

## Acknowledgements

The LUMES laboratory acknowledges support from the NCCR MUST. G.B. acknowledges financial support from the Swiss National Science Foundation (SNSF) through the Grant No. 200021_159219/1. E.P. acknowledges financial support from the SNSF through an Advanced Postdoc Mobility Grant No. (P300P2_158473). F.J.G.d.A. acknowledges support from the Spanish MINECO (MAT2017-88492-R and SEV2015-0522) and Cellex Foundation. We would like to acknowledge T.T.A. Lummen for the initial design of the plasmonic Fabry–Perot resonator, and C.J. Chiara and J.J. Carroll for insightful discussions regarding ultrafast control of nuclear excitations.

## Author contributions

G.M.V., I.M., G.B., K.W., E.P. and I.K. conducted experiments; R.J.L. and D.M. fabricated samples; F.J.G.d.A. developed theory and performed calculations; G.M.V., I.M., G.B., I.K, B.B., F.J.G.d.A. and F.C. interpreted results; F.C. and F.J.G.d.A. conceived and designed the research; all authors have contributed to writing the article, and read and approved the final manuscript.

## Additional information

**Competing interests:** The authors declare no competing interests.

