## [Peer Review File · Nature Communications]

Reviewers' comments:

Reviewer #1 (Remarks to the Author):

As I commented previously, the simultaneous energy-momentum measurement and its theoretical support deserve a publication in Nature Communications (although I still think that the double-foil configuration reported in Refs. 31-34 is more versatile because it allows generation, characterization, and quantum-state reconstruction of ultrashort electron pulses). The results including the coherent control will be of great interests to specialists in ultrafast electron microscopy. Most of my questions/comments were clearly addressed in the revised manuscript. Therefore, I recommend the publication in Nature Communications.

One comment: regarding my last question (3), the authors gave a value of temporal coherence, several tens of femtoseconds, in the revised manuscript. I strongly recommend the authors to quantitatively give the minimum and maximum temporal coherence (large error is fine) expected from the comparison of theoretical results and experimental results in Figs 4 and 5. This can be very important information to all the researchers who use electron microscopes.

Reviewer #3 (Remarks to the Author):

In the revised version of the manuscript, the authors largely addressed the main issues raised by the referees regarding (1) the novelty of their approach in comparison to previous PINEM-type experiments, (2) the coherence length of electron pulses and (3) the feasibility of the zeptosecond coherent control experiment.

Regarding (1) novelty of the approach:

- I generally agree with the author's argument that their work extends the current methodology of PINEM experiments, by providing a first indication of photon-resolved transverse electron scattering (Fig. 3f, albeit at a low data quality). Although otherwise advocated in the reply by the authors, also the energy exchange observed in "traditional" PINEM experiments can be considered as a longitudinal momentum transfer (see e.g. p. 15 of the manuscript), so that the statement "for the first time the quantized momentum exchange between light and electron beams is visualized" is only true for a transverse momentum exchange.
- The use of flat membranes as interaction mediators between free electrons and optical fields seems to be already long known in the community (e.g. already in 2014 in Ref.28) and should not be described as a new approach.
- I agree that the application of delayed optical pulse pairs is new in PINEM-type interactions and represents an intriguing addition to the field. This part of the manuscript together with the demonstration of transverse momentum exchange provides in my opinion a sufficient level of novelty for publication in Nature Communications.

Regarding (2) the coherence length of the electron pulses:

- The authors provide a convincing argument why the homogeneous linewidth of single electron wave functions needs to be considered instead of the inhomogeneous linewidth. However, one might argue to not call this the "longitudinal coherence length/time" of the electron pulse, as this may be reserved for an ensemble property (averaged over many electron pulses), equivalent to the concept of a transverse coherence length used earlier in the manuscript.

Regarding (3) the feasibility of the zeptosecond coherent control experiment:

- Could the authors please provide an estimate for the absorbed X-ray energy per illuminated area in the proposed multilayer?

Overall, I recommend publication of the manuscript if the remaining questions and comments are addressed by the authors.

Further questions/comments:

- On p.7, the authors state that the observed dependencies on the sample tilt and polarization direction exclude that effect can be derived by the classical electron deflection in an electromagnetic field. I agree that the quantized momentum transfer requires the electron to be described as a wave. But I would expect that the overall dependencies on sample tilt and polarization direction are rather similar in the classical and quantum mechanical description. Could the authors please comment on this.

- I suggest expanding the description of the double-pulse interaction in the theoretical methods section (p.15-16), as this represents one of the main theoretical advancements. Eq. 4-9 are similarly stated in previous works (as cited in the manuscript), but Eq. 9 is new and should be properly discussed. (If I followed the derivation correctly, the authors substitute the whole argument of the Bessel functions in Eq. 7 by a sum of two terms gaussians weighted by complex beta values, instead of just using a sum of two beta values). As a side-remark, Eq. 9 contains both contributions from trivial optical interference of the two pulses, as well as interference between components of the electron wave function. It would be helpful to discuss these different contributions and give a detailed description of the physical picture involved.

- I might have missed it: Does the manuscript contain a description on how the optical pulse duration is varied and how it is precisely characterized?

- The authors also (very briefly) discuss a coherent control experiment on a plasmonic resonator structure, as an example for experiments which are feasible in a two-pulse PINEM scheme. In particular, a response curve of the resonator with a spectral width of 200 meV is extracted. What determines this spectral width? Could the same response curve be obtained with single optical pulses with a tunable central wavelength?

- On p.9 the authors describe a "coherent constructive and destructive modulation of beta". Does beta refer to a single (complex) number or to a function $\beta(r)$?

- For the zeptosecond coherent control experiment, it is not clear to me, how the modulation of electron interference by zeptosecond time-delays gives access to intramolecular electron motions or nuclear processes as envisioned in the conclusion of the manuscript- especially for the two driving pulses being about 100 fs apart. Could the authors please explain this in more detail.

Minor points:

- I still believe that the signs of alpha and delta in Fig. S4 are at odds with Eq. S1. In order for beta approaching zero, the two sine terms in Eq. S1 need to have an opposite sign, which implies an opposite sign for alpha and theta (for small alpha and theta), contrary to the labeling in Fig. S4.

Reviewer #1 (Remarks to the Author):

As I commented previously, the simultaneous energy-momentum measurement and its theoretical support deserve a publication in Nature Communications (although I still think that the double-foil configuration reported in Refs. 31-34 is more versatile because it allows generation, characterization, and quantum-state reconstruction of ultrashort electron pulses). The results including the coherent control will be of great interests to specialists in ultrafast electron microscopy. Most of my questions/comments were clearly addressed in the revised manuscript. Therefore, I recommend the publication in Nature Communications.

We thank the reviewer for his/her time and contribution in reviewing our manuscript, and appreciate his/her overall positive evaluation. His/her comments surely contributed to improve the quality of our paper.

One comment: regarding my last question (3), the authors gave a value of temporal coherence, several tens of femtoseconds, in the revised manuscript. I strongly recommend the authors to quantitatively give the minimum and maximum temporal coherence (large error is fine) expected from the comparison of theoretical results and experimental results in Figs 4 and 5. This can be very important information to all the researchers who use electron microscopes.

Following the reviewer comment, we used the experimental map in Fig. 4 to tentatively extract a rough estimate of the longitudinal temporal coherence of the single-electron wave function. Consider the envelope of the intensity profile of the high-energy spectral sidebands as a function of the delay time between the two pulses, $\Delta_2 - \Delta_1$. This envelope could be interpreted as the tail of a Gaussian function centered at $\Delta_2 - \Delta_1 = 0$, and it would represent the homogeneous broadening of the electron pulse. The least-square fit of the data for sidebands at $\ell = 13, 14$ and 15 with a Gaussian function results in values of the longitudinal coherence that vary between 52 and 78 fs, with an uncertainty of about ± 10 fs. These values are in agreement with the estimate proposed in section S7 of the SI based on the uncertainty principle. In fact, considering that the energy bandwidth, ΔE , of the wave function of the photoemitted electrons would be mainly determined by the bandwidth of the excitation pulses, which is about 20 meV in our case, the longitudinal temporal coherence, $\xi_t \approx \hbar\sqrt{8\ln 2}/\Delta E$, would be around 77.5 fs.

Following the reviewer comment, this discussion has now been included in section S7 of the revised SI.

Reviewer #3 (Remarks to the Author):

In the revised version of the manuscript, the authors largely addressed the main issues raised by the referees regarding (1) the novelty of their approach in comparison to previous PINEM-type experiments, (2) the coherence length of electron pulses and (3) the feasibility of the zeptosecond coherent control experiment.

Regarding (1) novelty of the approach:

- I generally agree with the author's argument that their work extends the current methodology of PINEM experiments, by providing a first indication of photon-resolved transverse electron scattering (Fig. 3f, albeit at a low data quality). Although otherwise advocated in the reply by the authors, also the energy exchange observed in "traditional" PINEM experiments can be considered as a longitudinal momentum

transfer (see e.g. p. 15 of the manuscript), so that the statement “for the first time the quantized momentum exchange between light and electron beams is visualized” is only true for a transverse momentum exchange.

We thank the reviewer for this pertinent suggestion. We have revised the manuscript to insert the word “transverse” in the introductory paragraph at p. 4 and in the discussion at p. 8.

- The use of flat membranes as interaction mediators between free electrons and optical fields seems to be already long known in the community (e.g. already in 2014 in Ref.28) and should not be described as a new approach.

We took special care in avoiding such novelty claims. In fact, when discussing the use of semi-infinite light field to mediate the electron-light interaction, we only refer to this approach as “a more general method for controlling and manipulating the strength of electron-photon interaction” (see p. 3).

- I agree that the application of delayed optical pulse pairs is new in PINEM-type interactions and represents an intriguing addition to the field. This part of the manuscript together with the demonstration of transverse momentum exchange provides in my opinion a sufficient level of novelty for publication in Nature Communications.

We appreciate the positive evaluation from the reviewer and the recognition that our results would be an intriguing addition to the field.

Regarding (2) the coherence length of the electron pulses:

- The authors provide a convincing argument why the homogeneous linewidth of single electron wave functions needs to be considered instead of the inhomogeneous linewidth. However, one might argue to not call this the “longitudinal coherence length/time” of the electron pulse, as this may be reserved for an ensemble property (averaged over many electron pulses), equivalent to the concept of a transverse coherence length used earlier in the manuscript.

We agree that saying “longitudinal coherence of the electron pulse” might refer to an ensemble property. So, following the reviewer comment, in the revised manuscript we have now specified that we refer to the ‘intrinsic’ temporal coherence of the single-electron wave function, i.e. its homogeneous broadening.

Regarding (3) the feasibility of the zeptosecond coherent control experiment:

- Could the authors please provide an estimate for the absorbed X-ray energy per illuminated area in the proposed multilayer?

Considering that the proposed multilayer has a reflection coefficient $R = 0.35$ for a 777-eV x-ray beam, and that the absorption coefficient, μ , for gold and cobalt at this energy is around $1.3 \cdot 10^5 \text{ cm}^{-1}$ (<https://physics.nist.gov/PhysRefData/FFast/html/form.html>), the absorbed energy per volume will be: $f_{abs}^V = f_0(1 - R)\mu$, where $f_0 = 5 \text{ J/cm}^2$ is the incident fluence. In a single layer of gold or cobalt with thickness d ($d = 1 \text{ nm}$ for gold and $d = 1.6 \text{ nm}$ for cobalt), the absorbed energy per unit area would then be: $f_{abs}^A = f_0(1 - R)\mu d$, which becomes 42.2 mJ/cm^2 for the gold layer and 67.5 mJ/cm^2 for the cobalt layer. These values are smaller than the typical values of tenths of J/cm^2 for damage threshold in metals.

Following the reviewer comment, this discussion has now been included in section S9 of the revised SI.

Overall, I recommend publication of the manuscript if the remaining questions and comments are addressed by the authors.

We thank the reviewer for his/her time and contribution in reviewing our manuscript, and appreciate his/her overall positive evaluation. His/her comments surely contributed to improve the quality of our paper.

Further questions/comments:

- On p.7, the authors state that the observed dependencies on the sample tilt and polarization direction exclude that effect can be derived by the classical electron deflection in an electromagnetic field. I agree that the quantized momentum transfer requires the electron to be described as a wave. But I would expect that the overall dependencies on sample tilt and polarization direction are rather similar in the classical and quantum mechanical description. Could the authors please comment on this.

We thank the reviewer for this pertinent comment: indeed, momentum quantitation can only be explained by describing the electron as a wave, but both quantum and classical results are determined by the parameter beta, which therefore imprints a similar dependence in both. Moreover, the lateral Lorentz force experienced by the electrons in a classical picture is generally negligible in the case of excitation at optical photon energies and above, and the observation of a quantized exchange in the energy-momentum space is a clear indication that a quantum picture has to be adopted.

We have modified the text at p. 8 in order to reflect these ideas.

- I suggest expanding the description of the double-pulse interaction in the theoretical methods section (p.15-16), as this represents one of the main theoretical advancements. Eq. 4-9 are similarly stated in previous works (as cited in the manuscript), but Eq. 9 is new and should be properly discussed. (If I followed the derivation correctly, the authors substitute the whole argument of the Bessel functions in Eq. 7 by a sum of two terms gaussians weighted by complex beta values, instead of just using a sum of two beta values). As a side-remark, Eq. 9 contains both contributions from trivial optical interference of the two pulses, as well as interference between components of the electron wave function. It would be helpful to discuss these different contributions and give a detailed description of the physical picture involved.

Following the reviewer comment, in the Methods section of the revised manuscript we have included more details on the derivation of the two-pulse formula, as well as a detailed description of its physical interpretation.

- I might have missed it: Does the manuscript contain a description on how the optical pulse duration is varied and how it is precisely characterized?

To vary the duration of the optical pulse we modified the temporal chirp of the laser amplifier output using a pair of tunable glass prisms, whereas its characterization was performed by optical cross-correlation. Following the reviewer comment, in the Methods section of the revised manuscript we have specified these additional details.

- The authors also (very briefly) discuss a coherent control experiment on a plasmonic resonator structure, as an example for experiments which are feasible in a two-pulse PINEM scheme. In particular, a response

curve of the resonator with a spectral width of 200 meV is extracted. What determines this spectral width? Could the same response curve be obtained with single optical pulses with a tunable central wavelength?

Regarding the first question, the spectral response curve of the resonance is obtained as the Fourier transform of the time profile shown in Fig. 6c. Its FWHM is thus determined by three different contributions: i) the plasmon lifetime (giving a broadening generally smaller than 40 meV), ii) the laser linewidth (generally around 20 meV), and iii) the finite temporal window in which the experiment is conducted (in our case it is around 20 fs, which would correspond to a broadening of about 206 meV). For the current dataset, the latter aspect is the limiting factor. However, when increasing the temporal window of acquisition to values around 150 fs, the method can potentially be used as a time-domain spectro-microscopy measurement of the plasmon lifetime.

Regarding the second question, the answer is yes. The spectral response curve of the resonance can be also obtained by using a single optical pulse with a tunable wavelength, and actually we demonstrated this concept in a recent publication (see ACS Photonics 5, 759–764 (2018)). In this case, there is no limitation imposed by the temporal window of acquisition and the resolution is determined by the 20-meV laser linewidth. We have modified the text at p. 11 in order to include this remark.

- On p.9 the authors describe a “coherent constructive and destructive modulation of beta”. Does beta refer to a single (complex) number or to a function $\beta(r)$?

For the experiment discussed in the current manuscript, where a homogeneous Ag layer has been adopted, both the electric field of the light on the surface and the interaction strength β do not depend on the position (there is only a temporal dependence along the electron path, although the experiment only depends on the value of β after interaction with the sample). However, in the general case of a nanostructured surface, both the optical electric field and β would indeed depend on position.

- For the zeptosecond coherent control experiment, it is not clear to me, how the modulation of electron interference by zeptosecond time-delays gives access to intramolecular electron motions or nuclear processes as envisioned in the conclusion of the manuscript- especially for the two driving pulses being about 100 fs apart. Could the authors please explain this in more detail.

In order to better clarify how an electron wave-function modulated on the attosecond/zeptosecond scale can be adopted for accessing nuclear processes, we have introduced in the revised manuscript (see pages 12 and 13) further details on the design of a possible experiment for the coherent control of nuclear excitation and reaction. In fact, controlling nuclear phenomena via external parameters would open an extremely interesting perspective for energy production and for generation of radiation. Although speculative, this discussion is based on very recent experiments (see C. J. Chiara, J. J. Carroll, et al. Nature 554, pages 216–218 (2018)) and new concepts that we are actively discussing with Dr. Chiara and Dr. Carroll. We believe that the proposed experiment will stimulate an interesting interdisciplinary discussion.

Minor points:

- I still believe that the signs of α and δ in Fig. S4 are at odds with Eq. S1. In order for β approaching zero, the two sine terms in Eq. S1 need to have an opposite sign, which implies an opposite sign for α and θ (for small α and θ), contrary to the labeling in Fig. S4.

We thank the reviewer for this pertinent comment: in Eq. S1 the formula for $|\beta|$ at $\vartheta = 0$ was previously reported incorrectly. We have now included the correct equation in the revised version of the SI.

REVIEWERS' COMMENTS:

Reviewer #3 (Remarks to the Author):

Vanacore et al. properly addressed my previous comments and questions, and I recommend publication of the manuscript.

A final remark on the title: As already mentioned by one of the other referees, the use of the term "zeptosecond" in the title may be somewhat misleading, especially for an experimentally oriented manuscript. Extending the PINEM methodology to X-ray energies, and thereby to zeptosecond timescales, is only theoretically proposed at the end of the manuscript. Therefore, I suggest to not use the term "zeptosecond" in the title.

Reply to reviewers

Reviewer #3 (Remarks to the Author):

Vanacore et al. properly addressed my previous comments and questions, and I recommend publication of the manuscript.

A final remark on the title: As already mentioned by one of the other referees, the use of the term “zeptosecond” in the title may be somewhat misleading, especially for an experimentally oriented manuscript. Extending the PINEM methodology to X-ray energies, and thereby to zeptosecond timescales, is only theoretically proposed at the end of the manuscript. Therefore, I suggest to not use the term “zeptosecond” in the title.

We thank the reviewer for his/her time and contribution in reviewing our manuscript, and appreciate his/her overall positive evaluation. His/her comments surely contributed to improve the quality of our paper. Following his/her comment we have now modified the title of the paper by removing the term “zeptosecond”.